

# Ozone Profile Climatology for Remote Sensing Retrieval Algorithms

Kai Yang[1] and Xiong Liu[2]

[1]Department of Atmospheric and Oceanic Sciences, University Maryland, College Park, MD 20742, USA
[2]Harvard-Smithsonian Center for Astrophysics, Cambridge, MA 02138, USA

**Correspondence:** Kai Yang (kaiyang@umd.edu)

**Abstract.**

New ozone ($O_3$) profile climatologies are created from the Modern-Era Retrospective Analysis for Research and Applications version 2 (MERRA-2) $O_3$ record between 2005 and 2016, within the period of Aura Microwave Limb Sounder (MLS) and Aura Ozone Monitoring Instrument (OMI) assimilation. These two climatologies consist of local solar time, longitudinal (15°), and latitudinal (10°) dependent monthly mean $O_3$ profiles and the corresponding covariances, which are parameterized respectively by tropopause pressure and total $O_3$ column. They are validated through comparisons, which show good agreements with previous $O_3$ profile climatologies. Compared to a monthly zonal mean climatology, both tropopause- and column-dependent climatologies provide improved *a priori* information for profile and total $O_3$ retrievals from remote sensing measurements. Furthermore, parameterization of $O_3$ profile with total column usually reduces the natural variability of the resulting climatological profile in the upper stratosphere further than the tropopause parameterization, which usually performs better in the upper troposphere and lower stratosphere (UTLS). Therefore tropopause-dependent climatology is more appropriate for profile $O_3$ retrieval for complementing the vertical resolution of backscattered ultraviolet (UV) spectra, while the column-dependent climatology is more suited for use in total $O_3$ retrieval algorithms, with an advantage of complete profile specification without requiring ancillary information. Compared to previous column-dependent climatologies, the new MERRA-2 column-dependent climatology better captures the diurnal, seasonal, and spatial variations and dynamical changes of $O_3$ profiles with higher resolutions in $O_3$, latitude, longitude, and season. The new MERRA-2 climatologies contain first quantitative characterization of $O_3$ profile covariances, which facilitate a new approach to improve $O_3$ profiles using the most probable patterns of profile adjustments represented by the empirical orthogonal functions (EOFs) of the covariance matrices. The MERRA-2 daytime column-dependent climatology is used in the combo $O_3$ and $SO_2$ algorithm for retrieval from the Earth Polychromatic Imaging Camera (EPIC) onboard the Deep Space Climate Observatory (DSCOVR) satellite, the Ozone Mapping and Profiler Suite Nadir Mapper (OMPS-NM) on the Suomi National Polar Partnership (SNPP), and the Ozone Monitoring Instrument (OMI) on the Aura spacecraft.



## 1 Introduction

Remote sensing instruments measure spectral radiances, from which information about light absorbers such as ozone ($O_3$) and other trace gases, may be inferred using retrieval algorithms. Absorption signals carried in the measured radiances come from the interaction between photons and these absorbers, which are naturally distributed throughout the atmosphere. Consequently,

for a band within the spectral range of significant atmospheric absorptions, its measured radiance is sensitive to the profiles, i.e., vertical distributions of light absorbers, as well as depends on other atmospheric state and surface variables. However, multispectral or even hyperspectral radiance measurements from nadir viewing instruments do not have sufficient vertical resolution to fully disentangle the absorption signals for the determination of a absorber profile. Hence, the retrieval of quantitative information about an absorber requires some knowledge of its vertical distribution. Frequently, prescribed or *a priori* profiles

are used to fill the knowledge gap on the altitudes from which absorption signals are originated but not differentiated by the measurements.

For retrieval of $O_3$ from remote sensing measurements, the *a priori* knowledge is usually taken from an $O_3$ profile climatology, which provides average $O_3$ profiles and their variances. Most, if not all, total $O_3$ algorithms (e.g. Mateer et al., 1971; Klenk et al., 1982; Bhartia and Wellemeyer, 2002; Coldewey-Egbers et al., 2005; Eskes et al., 2005; Veefkind et al., 2006; Van

Roozendael et al., 2006; Lerot et al., 2010; Loyola et al., 2011; Van Roozendael et al., 2012; Lerot et al., 2014; Wassmann et al., 2015) rely on an $O_3$ profile climatology to uniquely and completely specify the vertical distribution of a retrieved total column. Profile $O_3$ algorithms (e.g. Hoogen et al., 1999; Liu et al., 2005; Wei et al., 2010; Bhartia et al., 2013; Miles et al., 2015) based on the widely adopted optimal estimation (OE) inversion technique (Rodgers, 2000) require the *a priori* $O_3$ profiles and their covariances to constrain the retrievals from deviating too far from the *a priori* $O_3$ distributions. An OE retrieved $O_3$ profile is a

combination, i.e., weighted average of the real and the *a priori* profiles. Therefore, the accuracy of a total column or a profile retrieval is significantly affected by the selection of the *a priori* profile. A closer match between the *a priori* and the actual vertical distributions allows a higher accuracy retrieval to be achieved.

In this paper, we outline the characteristics of $O_3$ vertical distribution and existing climatological data sets that are commonly used in many $O_3$ retrieval algorithms. To improve *a priori* knowledge of $O_3$ vertical distributions and its covariances and

to address the deficiencies in existing $O_3$ profile climatologies, we construct new $O_3$ profile climatologies from the long-term global $O_3$ field provided by a modern reanalysis system. We present comparisons to validate the new climatologies and summarize the present work in the last section.

## 2 Characteristics and Climatologies of $O_3$ Vertical Distribution

$O_3$ profile climatologies (e.g. Wellemeyer et al., 1997; Fortuin and Kelder, 1998; Bhartia and Wellemeyer, 2002; Lamsal et al.,

2004; McPeters et al., 2007; Wei et al., 2010; McPeters and Labow, 2012; Bak et al., 2013; Sofieva et al., 2014; Labow et al., 2015) usually provide the *a priori* profiles for $O_3$ retrieval from nadir-viewing satellite observations. Most of these climatologies are constructed by merging lower atmosphere ozonesonde data with upper atmosphere $O_3$ profile measurements from one or more satellite instruments, such as the Microwave Limb Sounder (MLS) and the Stratospheric Aerosol and Gas Experiment



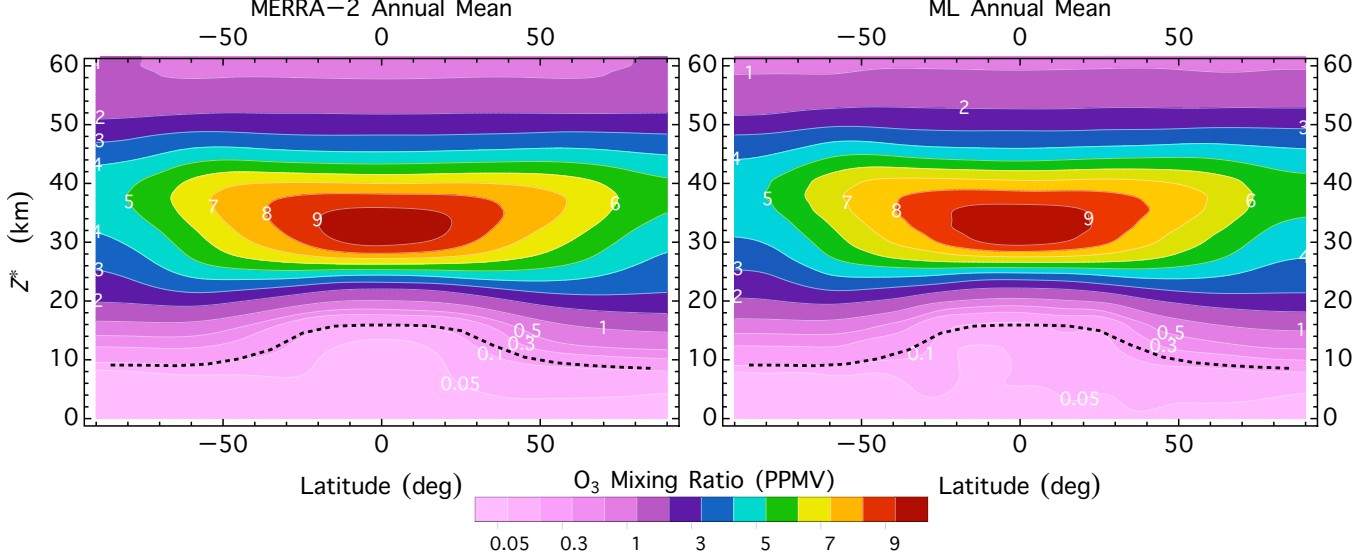

**Figure 1.** Annual zonal mean $O_3$ mixing ratio in ppmv as a function of latitude and pressure altitude $Z^*$. The left panel is from the new $O_3$ profile climatology described in this paper, and the right panel is from the ML climatology (McPeters and Labow, 2012). The dotted black line in each panel indicates the pressure altitude of the annual zonal mean tropopause. The pressure altitude $Z^*$ is defined as $16 \log_{10}\left[\frac{p_s}{p}\right]$, where $p$ is pressure level (in hPa) and $p_s = 1013.25$ hPa.

II (SAGE-II) on the Upper Atmosphere Research Satellite (UARS), the MLS on Aura, or the Solar Backscatter Ultraviolet (SBUV) and SBUV/2 on NASA and NOAA satellites. All these climatologies capture the main characteristic of $O_3$ vertical distribution, which is determined by the balance of the chemical processes of $O_3$ production and destruction, as well as by atmospheric motions. Specifically, the vertical $O_3$ distribution varies strongly with latitude, as shown in Fig. 1. The highest $O_3$

mixing ratio is found at an altitude of 30–40 km in the equatorial region (see Fig. 1), in which most atmospheric $O_3$ production induced by strong solar ultraviolet (UV) radiation takes place. The peak values of $O_3$ profiles in mixing ratio decrease with higher latitudes, as the large-scale meridional Brewer-Dobson circulation carries the $O_3$ in the tropical stratosphere towards the poles, and slowly transfers the $O_3$ to the lower stratosphere at middle and high latitudes. As a result of this atmospheric circulation driven process of transport, descent, and accumulation, $O_3$ profiles in partial pressure or number density exhibit

higher maximum values occurring at lower altitudes as latitude increases towards the poles (see Fig. 14). In addition to latitude dependence, some climatologies (e.g. Fortuin and Kelder, 1998; McPeters et al., 2007; Wei et al., 2010; McPeters and Labow, 2012; Bak et al., 2013; Sofieva et al., 2014) include monthly zonal mean $O_3$ profiles to describe systematic profile changes caused by the seasonal variations in ozone photochemistry and atmospheric circulation, as well as significant hemispheric profile asymmetry resulting from hemispheric differences in orography, atmospheric temperature, and circulation transport (Maeda

and Heath, 1983; Perliski and London, 1989; Cariolle et al., 1992). While the seasonal dependence is accounted for in these



$O_3$ profile climatologies, they have not included the diurnal cycle of $O_3$, which is significant in the upper stratosphere and mesosphere (Haefele et al., 2008; Huang et al., 2008; Sakazaki et al., 2013; Schranz et al., 2018).

Tropopause marks the interface between the stratosphere and the troposphere, across which there is a steep vertical gradient in $O_3$ mixing ratio, which is higher in the stratosphere than in the troposphere (see Fig. 1). In the lower stratosphere (between the tropopause and ~30 km altitude) where $O_3$ lifetime is quite long (on the order of weeks or longer), the deviation of $O_3$ vertical distribution from its climatological mean is mainly controlled by atmospheric dynamics. Consequently, $O_3$ profile and column amount have large daily variations that are associated with meteorological conditions (Reed, 1950). Especially the rise and fall of tropopause affect $O_3$ columns and its vertical profiles directly through shifting the $O_3$ mixing ratio gradient in the upper troposphere and the lower stratosphere (UTLS). This dynamical connection between total $O_3$ and tropopause pressure (or altitude) have been investigated and documented in a number of studies (e.g. Ohring and Muench, 1960; Salby and Callaghan, 1993; Steinbrecht et al., 1998; Krzyścin et al., 1998; Weiss et al., 2001; Varotsos et al., 2004), which reveal a strong positive (negative) correlation between total $O_3$ and tropopause pressure (altitude). This correlation implies that tropopause height (or pressure) and total $O_3$ column amount are excellent indicators for selecting $O_3$ profile shape in the UTLS region, where $O_3$ profiles have the highest dynamical variability. To capture dynamical profile variations, $O_3$ profile climatologies have included tropopause-sensitive zonal mean $O_3$ profiles (Wei et al., 2010; Bak et al., 2013; Sofieva et al., 2014), as well as column classifications for which zonal mean $O_3$ profiles are compiled for a range of possible total $O_3$ columns (Wellemeyer et al., 1997; Bhartia and Wellemeyer, 2002; Lamsal et al., 2004; Labow et al., 2015).

The accuracy of the global $O_3$ measurements from the series of Total Ozone Mapping Spectrometer (TOMS) owes in large part to the *a priori* knowledge of $O_3$ vertical distribution provided by the total-ozone-column-classified $O_3$ climatology (Bhartia and Wellemeyer, 2002; McPeters et al., 2007) created for the version 8 (V8) total $O_3$ algorithm. For mapping between $O_3$ column and profile, many recent total $O_3$ algorithms (e.g. Bhartia and Wellemeyer, 2002; Eskes et al., 2005; Veefkind et al., 2006; Van Roozendael et al., 2006; Lerot et al., 2010; Loyola et al., 2011; Van Roozendael et al., 2012; Lerot et al., 2014; Wassmann et al., 2015) use the TOMS-V8 climatology, which is a combination of latitude and total $O_3$ dependent profiles, also known as the standard profiles (Bhartia and Wellemeyer, 2002) and the latitude and month dependent Labow-Logan-McPeters (LLM) climatology (McPeters et al., 2007). The standard profiles are twenty-one annual mean $O_3$ profiles, covering the possible total $O_3$ ranges in steps of 50 DU for three latitude zones: 225–325 DU at low-latitudes, 225–575 DU at mid-latitudes, and 125–575 DU at high latitudes, and they are used to expand the latitude and month dependent LLM climatology to include the variation with total $O_3$. While the LLM climatology provides latitude-dependent $O_3$ profiles that capture the north-south asymmetry, the profile dependence on total $O_3$ taken from the standard profiles is independent of season and makes no distinction between northern and southern hemispheres, therefore ignore the hemispheric asymmetry in $O_3$ profile deviation from the climatological mean. Furthermore, profile changes associated with total column variations represented by the standard profiles exhibit large morphology difference between two adjacent latitude zones, as a result of these standard profiles are binned over wide (30°) latitude zones. Consequently, total $O_3$ algorithms relying on TOMS-V8 $O_3$ profile climatology for profile specification usually have systematic latitudinal biases in retrieved $O_3$ columns due to the use of hemispherically symmetric and time-independent profile adjustments, and $O_3$ column discontinuities across the latitude zone boundaries due to large morphology differences.





The deficiency of symmetric profile changes may be addressed with the newer column-dependent $O_3$ climatologies with hemispheric separation (Lamsal et al., 2004; Labow et al., 2015), but their latitude dependencies are compiled on the same 30° wide latitude zones as the standard profiles, with a coarse (semi-annual) seasonal variation (Lamsal et al., 2004) or without seasonal distinction (Labow et al., 2015).

Since the TOMS-V8 climatology is widely used today by many different total $O_3$ algorithms, there is a need to eliminate its deficiencies by creating a new $O_3$ profile climatology. The primary objective of the investigation presented in this paper is to develop a new column-classified $O_3$ climatology using recent data to provide realistic *a priori* $O_3$ profile specifications in total $O_3$ retrievals from the ultraviolet (UV) observations from the Earth Polychromatic Imaging Camera (EPIC) onboard the Deep Space Climate Observatory (DSCOVR) satellite, the Ozone Mapping and Profiler Suite Nadir Mapper (OMPS-NM) on

the Suomi National Polar Partnership (SNPP) and NOAA-20 satellites, the Ozone Monitoring Instrument (OMI) on Aura, or any other instruments on current and future satellites.

Profile $O_3$ retrieval algorithms based on the optimal inversion method (Rodgers, 2000) need not only *a priori* $O_3$ profiles but also the associated profile covariance matrices to constrain the retrieved profiles. However, $O_3$ climatologies (e.g. McPeters et al., 2007; McPeters and Labow, 2012; Bak et al., 2013; Sofieva et al., 2014) that are frequently employed in profile retrievals

do not contain information about $O_3$ profile covariance, instead the needed covariance matrices are constructed by assuming positive covariance between different atmospheric layers (e.g. Hoogen et al., 1999; Liu et al., 2005; Bhartia et al., 2013; Miles et al., 2015). This assumption is often unrealistic, as $O_3$ profile changes resulting from atmospheric vertical motions usually have negative correlations among layers in the UTLS. Improved knowledge about the variability of $O_3$ vertical distributions, especially how changes among different layers are related, benefit both profile and total column retrievals. Therefore, it is

important to develop a new $O_3$ profile climatology that provides quantitative information about $O_3$ profile covariance.

## 3   MERRA-2 $O_3$ Field

The Modern-Era Retrospective Analysis for Research and Applications version 2 (MERRA-2, Bosilovich et al. 2015; Gelaro et al. 2017) reanalysis project produces assimilated data products, including meteorological fields (such as atmospheric temperature and tropopause pressure and attitude) and an $O_3$ field. This global $O_3$ field is driven by atmospheric dynamics and

constrained by satellite $O_3$ measurements and is continuous in time and three-dimensional space. Beginning in October 2004, Aura MLS provides constraints on $O_3$ profile from the lower mesosphere to the upper troposphere. Though MLS does not have information from the lower troposphere, Aura OMI and MLS jointly provide constraints on the tropospheric $O_3$ column, with its vertical distribution controlled by atmospheric transport and simplified chemistry with parameterized $O_3$ loss implemented in the MERRA-2 assimilation system (Wargan et al., 2015; Bosilovich et al., 2015). Evaluation of MERRA-2 $O_3$ shows excel-

lent agreement between the assimilated $O_3$ profile in the stratosphere and upper troposphere (between 1 hPa and 500 hPa) with independent satellite and ozonesonde measurements, and that MERRA-2 assimilation reproduces realistically the stratospheric and upper tropospheric $O_3$ variability (Wargan et al., 2015, 2017). In the lower troposphere, the correlations between MEERA-2 $O_3$ profiles and the coincident ozonesonde measurements are lower than the correlations in the UTLS  (Wargan et al., 2015,



2017). But the tropospheric $O_3$ columns (which are $O_3$ amounts resulting from surface-to-tropopause profile integration) agree very well and have a high degree of correlation with the corresponding columns from ozonesonde measurements (Ziemke et al., 2014), validating the integrated tropospheric profiles of MERRA-2. In short, MERRA-2 assimilated $O_3$ field provides a realistic representation of atmospheric $O_3$ that faithfully captures both short-term (daily or shorter) variations and seasonal changes

in vertical and horizontal distributions and thus contains vast information to be harvested for improved characterization of $O_3$ vertical distributions and variations.

## 4   MERRA-2 $O_3$ and Temperature Profile Climatologies

The MERRA-2 record since the Aura MLS and OMI assimilation, owing to its realistic representation of atmospheric $O_3$, is suitable for creating climatologies to provide improved knowledge of $O_3$ and temperature vertical distributions.

The long-term MERRA-2 record is stored as a four-dimensional (latitude, longitude, atmospheric pressure, and time) data set that covers the globe uniformly at high spatial and temporal resolutions. More precisely, the global coverage is provided at 3-hourly intervals with a horizontal resolution of 0.5° latitude by 0.625° longitude and a vertical grid covering 72 layers between the surface and 0.01 hPa. The MERRA-2 $O_3$ profiles from 2005 to 2016, within the period of Aura MLS and OMI assimilation, are analyzed to create a new $O_3$ profile climatology. The enormous amount of data from this period facilitate

reliable statistical representation of mean $O_3$ profiles and their variations on more finely resolved dependent variables, including tropopause pressure, total column amount, latitude, longitude, and time. Note that MERRA-2 data fields are on a hybrid terrain-following pressure coordinate (Wargan et al., 2017), and are interpolated to the uniform pressure altitude grid before statistical computation to create MERRA-2 climatologies. Here the pressure altitude, $Z^* = 16 \log_{10}[\frac{p_s}{p}]$, is a dimensionless quantity, but is assigned units of km, because it is quite close to the altitude (in km) at pressure level $p$ above the surface at $p_s$=1013.25 hPa,

with a difference usually less than 1 km when $Z^* \lesssim 30$ km.

### 4.1   Local-Solar-Time-Dependent $O_3$ Profile Climatology

The photochemical $O_3$ production and destruction depend strongly on solar illumination, which changes systematically for a location during the course of a day, resulting in diurnal variation of $O_3$ vertical profiles (Haefele et al., 2008; Huang et al., 2008; Sakazaki et al., 2013; Schranz et al., 2018), but existing $O_3$ profile climatologies have not included $O_3$ profile dependence on

local solar time. Remote sensing $O_3$ measurements are collected at different local solar times, even from the same instrument, such as the DSCOVR EPIC, which observes the Earth from sunrise to sunset simultaneously (Herman et al., 2017). Proper accounting for the diurnal variations in $O_3$ vertical distribution would enable more accurate $O_3$ retrieval from remote sensing observations. Therefore we construct a local-solar-time-dependent $O_3$ profile climatology from the MERRA-2 $O_3$ field.

   We divide the globe equally into 24×18 rectangle tiles, and each has the size of 15°-longitude × 10°-latitude. Monthly mean

profiles and their variances are calculated from samples that fall within a tile at the eight UTC times of MERRA-2 $O_3$ field. Note that data samples are weighted by the cosine of latitude in performing the statistics to ensure equal weighting by area.





Since each climatological profile of a tile is compiled from $O_3$ profiles at the same UTC, it represents an hourly mean because a tile's 15° longitude range is equivalent to one hour of local solar time (LST).

**Figure 2.** MERRA-2 monthly mean $O_3$ profiles for four months (January, April, July, and October) and seven tiles that cover the same longitude range (0°E-15°E) and seven different latitude zones (90°S-80°S, 50°S-40°S, 30°S-20°S, 00°N-10°N, 20°N-30°N, 40°N-50°N, and 80°N-90°N). Each profile (plotted as a colored line) is calculated from samples with the same UTC time, with different colors represent different UTC times shown in the legend. These mean profiles are averages over 15° longitude, which is equivalent to one hour of local solar time (LST).





Figure 2 shows the sample $O_3$ profiles from the MERRA-2 local-solar-time-dependent climatology, illustrating the diurnal, seasonal, and latitudinal variations of $O_3$ vertical distributions. Each panel in Fig. 2 contains plots of eight climatological $O_3$ mixing ratio profiles for a tile next to and east of the prime meridian, respectively for eight different one-hour LST periods. The differences among the curves in a panel reveal the diurnal $O_3$ mixing ratio changes, which in general increase significantly

with altitude from the upper ($Z^* \gtrsim 30$ km) stratosphere into the mesosphere ($Z^* \gtrsim 50$ km). The $O_3$ mixing ratio level in the mesosphere reaches its minimum after sunrise but then increases after sunset, recovering its night-time value quickly, while in the stratosphere where the peak value of $O_3$ mixing ratio is located ($Z^*$ between $\sim$35 – 40 km), $O_3$ diurnal cycle has its maximum in the afternoon and a lower value close to its minimum in the morning. The diurnal variations, which are mainly driven by photochemistry in the upper atmosphere, exhibit a strong latitudinal (panel rows in Fig. 2) and seasonal

(panel columns) dependence, as expected from the variations in solar insolation with the latitude and the season. The mean nighttime $O_3$ mixing ratio can be an order of magnitude higher than the daytime value in the mesosphere at $Z^* = 70$ km. In general, the magnitude of diurnal difference reduces with the altitude, with a maximum of $\sim$7% near the stratospheric $O_3$ mixing ratio peak. These significant diurnal $O_3$ mixing ratio variations correspond to typically 0.5% and a maximum of $\sim$4% peak-to-trough differences in diurnal variation of total $O_3$ vertical column. The results for the mesosphere and stratosphere

are in general consistent with previous findings of $O_3$ diurnal variations (Haefele et al., 2008; Huang et al., 2008; Sakazaki et al., 2013; Schranz et al., 2018, and references therein). Thus, this local-solar-time-dependent climatology properly captures diurnal variation in $O_3$ vertical distributions.

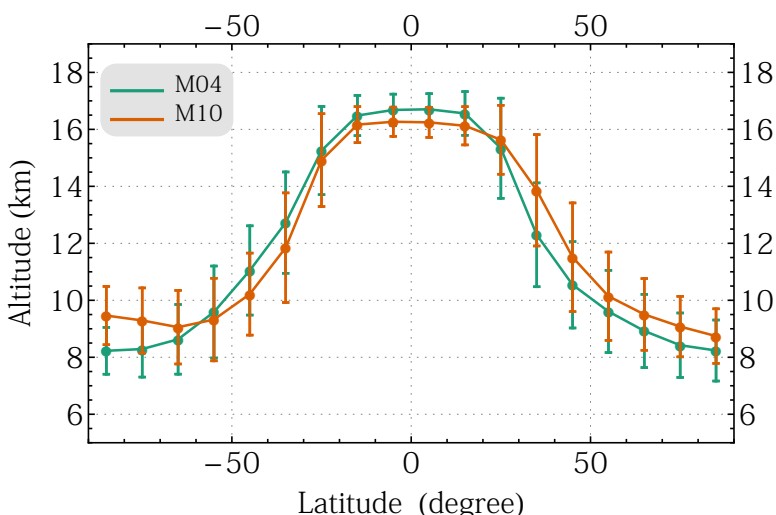

**Figure 3.** MERRA-2 daytime monthly zonal mean tropopause altitude and its standard deviation as a function of latitude for April (M04) and October (M10).





**Figure 4.** Profile comparisons between M2TPO3 and TpO$_3$ of Sofieva et al. (2014) for four months (January, April, July, and October) and seven latitude zones (90°S-80°S, 50°S-40°S, 30°S-20°S, 00°N-10°N, 20°N-30°N, 40°N-50°N, and 80°N-90°N). Colored solid lines represent M2TPO3 profiles, while the dotted ones for TpO$_3$ profiles. The color of a solid line indicates the percentage occurrence of the profile, which is calculated as the percentage of profiles in the month-latitude class that fall within the $Z^*$ tropopause pressure bin. The line legends display the average tropopause altitude and the average total O$_3$ column for the corresponding climatological profile. The solid gray line represents the downgraded M2TPO3 profile, i.e., the monthly zonal mean profile.





**Figure 5.** Similar to Fig. 4, but for standard deviation (SD) comparisons between M2TPO3 and TpO$_3$ of Sofieva et al. (2014).

## 4.2 Tropopause-Pressure-Classified O$_3$ Profile (M2TPO3) Climatology

Tropopause altitude varies with month and latitude systematically (Hoinka, 1998) and is usually highest in the tropics and drops toward the poles with steep declines in the mid-latitudes but exhibits hemispheric asymmetry (e.g., see Fig. 3). This figure also illustrates that the dynamic variability of tropopause altitude is characterized by daily standard deviations of approximately 1–2




km, which agrees well with the characterization by Seidel and Randel (2006). Since $O_3$ concentration is highest in the lower stratosphere above the tropopause and decreases rapidly in the troposphere, the large short-term fluctuation of tropopause altitude results in high variability in total $O_3$ column, accounting for $\sim$60% of its daily variation (Wei et al., 2010). Thus grouping $O_3$ profiles according to the tropopause pressure or altitude generates climatological profiles that reflect the dynamical

influences on $O_3$ vertical distributions, as demonstrated by the tropopause-altitude-classified $O_3$ profile climatology (named the TpO$_3$ climatology) created by Sofieva et al. (2014) using ozonesonde and SAGE-II data.

Similarly, we create a tropopause-pressure-classified $O_3$ profile climatology (referred to as M2TPO3 climatology hereafter) using the MERRA-2 $O_3$ record, which provides a large amount of data covering a diverse range of conditions, likely including all possible tropopause pressures. These mean profiles and their variances are calculated by statistically analyzing the sets of

MERRA-2 $O_3$ profiles at the same UTC time, binned by tropopause pressure in 1-km pressure altitude ($Z^*$) steps, calendar month, and $15° \times 10°$ longitude-latitude tile. To compare the M2TPO3 with the TpO$_3$ (Sofieva et al., 2014), we create a daytime zonal mean climatology by combining M2TPO3 tiles with the same latitude zone and LST from 9 AM through 5 PM. The resulting daytime climatology contains 2154 mean profiles and standard deviations (shown in Appendix A), spanning the range of tropopuase pressure (altitude) from 605 hPa (3.56 km) to 62 hPa (19.3 km), distributing in twelve calendar months

and eighteen 10° latitude zones that cover the latitude range from -90° to 90°.

Figures 4 and 5 display a subset of daytime M2TPO3 profiles and standard deviations and their comparisons with those of TpO$_3$(Sofieva et al., 2014). The results in Fig. 4 show that both M2TPO3 and TpO$_3$ have similar $O_3$ profile shapes for similar tropopause altitudes in each month-latitude class. Especially for the profiles in the tropics, where M2TPO3 and TpO$_3$ show very close profiles, that is nearly independent of tropopause altitude and season. For profiles at higher latitudes, Fig. 4

illustrates that the profile change associated with tropopause altitude variation occurs mostly below the $O_3$ concentration peak, exhibiting increasing UTLS $O_3$ with lower tropopause altitude. The $O_3$ column amounts (displayed in the figure legends) of TpO$_3$ profiles are generally (about 1 to 4%) higher than the corresponding M2TPO3 profiles. This comparison indicates that there is likely a small and high bias in the total columns in the TpO$_3$ climatology, since MERRA-2 total $O_3$ columns have a latitude-dependent (up to 2%) low biases (Wargan et al., 2017). Comparing with the TpO$_3$, there are more tropopause

pressure bins with sufficient samples for reliable statistics, hence more M2TPO3 profiles corresponding to a broader range of tropopause altitudes in each month-latitude class. Under the $O_3$ hole condition, strong $O_3$ spatial variations sampled differently by ozonesondes likely contributes to significant profile differences between M2TPO3 and TpO$_3$ (see Fig. 4 lower left panel).

The standard deviations in Fig. 5 show similar magnitudes and vertical structures between the M2TPO3 and the TpO$_3$ profiles with similar tropopause altitudes, demonstrating that the M2TPO3 climatology represents $O_3$ variability realistically.

The main differences occur in the lower troposphere of tropical latitude zones, where TpO$_3$ shows higher variability than the M2TPO3, due to ozonesondes better capture the influence from tropospheric $O_3$ productions, which are not explicitly included in the MERRA-2 reanalysis. Significant differences are also observed over polar latitude zones (90°S-80°S and 80°N-90°S), likely due to differences in sampling over highly inhomogeneous spatial distributions in these zones.





Results in Fig. 5 show that the standard deviations of the high occurrence profiles are in general smaller than those for the downgraded (monthly zonal mean) profile, indicating that tropopause pressure is a good predictor of $O_3$ profile shape, leading to a more accurate profile specification than simply taking the monthly zonal mean profile.

The comparison in this section shows overall good agreement of the M2TPO3 means and standard deviations versus those

of the TpO$_3$, with an expected discrepancy in the $O_3$ variability in the tropical lower troposphere, validating the realism of MEERA-2 $O_3$ record and its suitability for climatology construction.

### 4.3 Total-Column-Classified $O_3$ Profile (M2TCO3) Climatology

The tropopause-pressure dependent $O_3$ profiles contained in the M2TPO3 climatology capture profile changes resulting from short-term meteorological disturbances in the UTLS region. The near linear relationship between mean total column $O_3$ and

tropopause altitude in (e.g., see legend tables in Fig. 4) quantified for each month-latitude class implies that the total column $O_3$ may serve as a good predictor of $O_3$ profile shape as well. Grouping profiles by total column $O_3$ may actually better capture $O_3$ distribution and variability driven by both dynamical and chemical processes, we therefore create a total-column-classified $O_3$ profile climatology (referred to as the M2TCO3 climatology hereafter) from the long-term MERRA-2 $O_3$ record.

The M2TCO3 climatology is constructed by binning the MERRA-2 $O_3$ field at the same UTC time by total column in 25-

DU steps, calendar month, and $15° \times 10°$ longitude-latitude tile. To compare the M2TCO3 with the TOMS-V8 (Bhartia and Wellemeyer, 2002; McPeters et al., 2007), we create a daytime zonal mean climatology by combining M2TCO3 tiles with the same latitude zone and LST from 9 AM through 5 PM. The resulting climatology contains 1644 mean profiles and standard deviations (shown in Appendix A), spanning the range of total $O_3$ from 94 to 584 DU, distributing in twelve calendar months and eighteen latitude zones that cover the latitude range from -90° to 90°. Figure 6 displays a subset of daytime M2TCO3

profiles and the corresponding TOMS-V8 profiles, illustrating the distinct characteristics of $O_3$ vertical distributions, as well as significant differences between the two climatologies. Both M2TCO3 and TOMS-V8 profiles exhibit higher altitudes of $O_3$ peak with lower $O_3$ columns for a month-latitude class, lower peak altitudes with higher latitudes for a total $O_3$ column, narrow dynamical $O_3$ ranges and nearly identical profile shapes for different seasons in the tropics but higher seasonal and dynamical variations in mid- and high-latitude regions, and hemispherical asymmetry.

The differences between M2TCO3 and TOMS-V8 (shown in each panel of Fig. 6) reflect the improvements in the realism of $O_3$ profile representation with the M2TCO3 climatology. The column-averaged TOMS-V8 profiles are represented by the LLM climatology (McPeters et al., 2007), which is updated to the ML climatology (McPeters and Labow, 2012), and are very close to the downgraded M2TCO3 profiles (see comparisons in Fig. 1 and Fig. 14). Therefore the differences between M2TCO3 and TOMS-V8 profiles shown in Fig. 6 are attributed to the profiles deviations applied to the mean derived from the

standard column-dependent profiles of TOMS-V8 (Bhartia and Wellemeyer, 2002), which are independent of season, coarse in latitude resolution, and symmetric with respect to the equator. These deficiencies limit the realism of TOMS-V8 climatological profiles, and are eliminated with the new M2TCO3 climatology, which provides improved $O_3$ profile representation that better captures profile changes associated with column variations and their dependence on season and latitude.





**Figure 6.** Profile comparisons between M2TCO3 and TOMS-V8 for four months and seven latitude zones (same as those in Fig. 4). Colored solid lines represent M2TCO3 profiles, while the dotted ones for TOMS-V8 profiles. The color of a solid line indicates the percentage occurrence of the climatological profile, and its line legend displays the average tropopause altitude and the average total column O$_3$. The solid gray line represents the downgraded M2TCO3 (monthly zonal mean) profile, which is the same as the downgraded M2TPO3 profile shown in Fig. 4.

Figure 7 shows the standard deviations of a subset of M2TCO3 profiles with high ($> 5\%$) occurrence percentage and comparisons with those of M2TPO3. A vast majority of high occurrence climatological profiles of M2TCO3 and M2TPO3 shown





**Figure 7.** Standard deviation (SD) comparisons between M2TPO3 and M2TCO3 for four months and seven latitude zones (same as those in Fig. 4). Colored solid lines represent M2TCO3 SDs, while colored dotted lines for M2TPO3 SDs. The color of a solid or dotted line indicates the percentage occurrence of the climatological profile, and its line legend displays the average tropopause altitude and the average total column $O_3$. Only SDs for climatological profile with greater than 5% occurrence are plotted. The solid gray line represents the downgraded M2TCO3 profile SD, i.e., the SD for the monthly zonal mean profile, and it is identical to the SD of downgraded M2TPO3 profile. The SDs quantify the natural variability of $O_3$, and a smaller one signifies the climatological mean provides a more reliable $O_3$ profile specification.



in Fig. 7 exhibit significantly reduced standard deviations compared to those of monthly zonal mean (i.e., downgraded) profiles, illustrating that both tropopause pressure and total column $O_3$ provide information for more precise specification of $O_3$ profiles. Comparisons between M2TPO3 and M2TCO3 show that column classification usually leads to greater reductions in standard deviations in the upper stratosphere but smaller ones in the UTLS than tropopause pressure classification. Since there

are much more $O_3$ in the upper stratosphere than below, an overall more realistic specification of $O_3$ profile is achieved using the M2TCO3 climatology based on the total column $O_3$ than using the M2TPO3 based on the tropopause pressure. Hence the column-dependent climatology is most appropriate in mapping between $O_3$ column and vertical profile needed in total $O_3$ column retrieval algorithms. However, with the knowledge of tropopause pressure, the tropopause-dependent climatology usually provides closer matches to actual profiles and stronger constraints in the UTLS, therefore its usage significantly improves the

accuracy $O_3$ profile retrieval (Bak et al., 2013) from backscattered UV measurements, which have lower vertical resolutions in the troposphere than in the stratosphere (Liu et al., 2010).

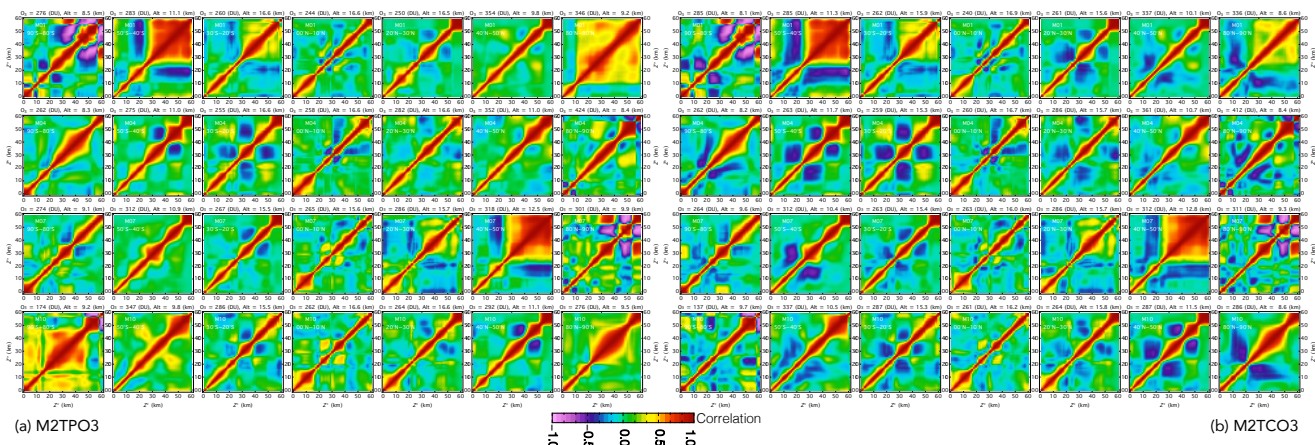

(a) M2TPO3       Correlation       (b) M2TCO3

**Figure 8.** Correlation matrices from M2TPO3 (left panels) and M2TCO3 (right panels) for four months and seven latitude zones (same as those in Fig. 4). Here the matrix for the highest occurrence climatological profile is shown for each month-latitude class.

## 4.4 $O_3$ Profile Covariance Climatology

Previous $O_3$ climatologies (e.g. Fortuin and Kelder, 1998; McPeters et al., 2007; McPeters and Labow, 2012; Bak et al., 2013; Sofieva et al., 2014) used in $O_3$ profile retrievals include profile standard deviations but not information about $O_3$ profile

covariance, because their sources (ozonesonde and satellite measurements) have limited coincident samples to quantify the joint $O_3$ variability between different altitudes throughout the atmosphere. In contrast, the MERRA-2 assimilation provides complete profiles simultaneously, allowing direct statistical computation of profile covariance matrices. We expand the classification analysis to quantify the $O_3$ profile covariance. Specifically, we construct an $O_3$ profile covariance matrix from each bin used for the calculation of an M2TPO3 or M2TCO3 climatological profile. The resulting covariance climatology provides





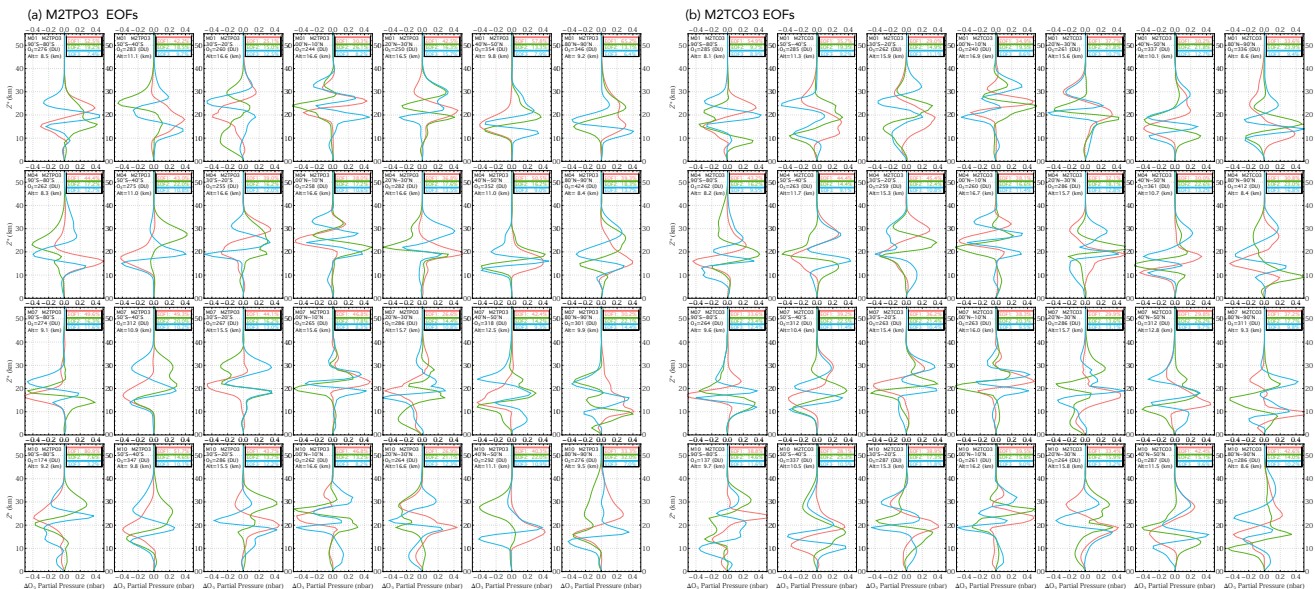

**Figure 9.** The empirical orthogonal functions (EOFs) of covariance matrices shown in Fig. 8. Only the first three ordered EOFs are plotted, with percentages of the class variance explained by the EOFs are displayed with the corresponding line legends.

first quantification of $O_3$ vertical distribution variabilities, the correlations among different levels, and their dependence on tropopause pressure or total column $O_3$ for different months and longitude-latitude tiles.

Figure 8 shows a subset of correlation matrices, which are standardized (i.e, diagonal element normalized to 1) covariance matrices, from the daytime M2TPO3 and daytime M2TCO3 climatologies. These density plots highlight the varying degree

of level-to-level correlations and their contrasts with the diagonal-constant matrices typically used in $O_3$ profile retrievals (e.g. Hoogen et al., 1999; Liu et al., 2005; Miles et al., 2015), which assume positive layer-to-layer correlation that decreases monotonically and exponentially with distance between layers. Contrary to this typical assumption, Fig. 8 illustrates that the degree of correlation between two levels fluctuates with the level separation and negative correlations are quite common.

The quantification of $O_3$ profile covariance offers a new way to represent $O_3$ vertical distribution realistically and efficiently.

Figure 9 shows the first three leading empirical orthogonal functions (EOFs), which are the ordered set of eigenvectors for each covariance matrix shown in Fig. 8. Typically the first three leading EOFs combined explain between 65% to 85% of the class variance, and the first fifteen leading EOFs accounts for over 95% of the class variance. Much like a climatological profile describes the likely $O_3$ vertical distribution, the EOFs describe the most probable patterns of profile deviations from the climatological mean. In general, an $O_3$ profile $\mathbf{X}$ can be expressed as

$$\mathbf{X} = \mathbf{X_m} + \sum_{k=1}^{n} \Omega_k \, \mathbf{e}_k, \tag{1}$$

where $\mathbf{X_m}$ is a climatological mean profile, and $\mathbf{e}_k$ the $k^{\text{th}}$ EOF, and $\Omega_k$ the $k^{\text{th}}$ coefficient, and $n$ the total number of EOFs with its maximum limited to the number of levels used to represent the $O_3$ profile. Since a high fraction of class variance may





be explained with just a few leading EOFs, they provide the most efficient adjustments to improve the representation of $O_3$ profile when it deviates from the climatological mean. When interpreted physically, the leading EOFs correspond to processes of $O_3$ accumulation, reduction, or redistribution. For instance, the first three EOFs for an M2TPO3 covariance matrix (see Fig. 9a) respectively describes 1) column increase or decrease, represented by the EOF-1 (monopole) pattern, 2) up or down

shift of a profile, represented by the EOF-2 (dipole) pattern, showing $O_3$ increase at one altitude while decrease at another, and 3) shrink or stretch of a profile, represented by the EOF-3 (tripole) pattern, showing $O_3$ decrease (increase) at one altitude and increase (decrease) at adjacent (above and below) altitudes. The results in Fig. 9a indicate that the highest or the second highest contribution to the variance of a tropopause class originates from the total column variation. However, the highest contribution to the variance of a total column class comes from $O_3$ profile shift, represented by EOF-1 (dipole) in Fig. 9b, linked usually

with tropopause variation, since the dominant profile change resulting from column variation is accounted for with different $O_3$ column classes. The second EOF (tripole) of an M2TCO3 covariance matrix describes the shrink or stretch of a profile, similar to the third EOF of an M2TPO3 matrix, while subsequent EOFs describe more complex rearrangement of $O_3$ profile.

## 4.5 Temperature Profile Climatology

Using the same binning schemes employed in the $O_3$ profile climatologies, we create temperature profile climatologies from the

MERRA-2 assimilated atmospheric temperature field. The resulting climatologies contain mean and covariance of temperature profiles, paired with the respective M2TPO3 and the M2TCO3 climatological $O_3$ profiles.

Figure 10 shows sample MERRA-2 climatological temperature profiles corresponding to the sample M2TCO3 climatological $O_3$ profiles shown in Fig. 6, and the $O_3$-temperature correlation profiles for the sample month-latitude classes. These results illustrate the systematic behavior among $O_3$ column amount, tropopause altitude, and atmospheric temperature: higher

$O_3$ column amounts occur with a warmer lower stratosphere and a colder troposphere, the condition for lower tropopause altitudes. This relationship, as well as its dependence on season and latitude, is captured in the MERRA-2 $O_3$ and temperature climatologies and is consistent with the findings of long-term $O_3$ and temperature profile measurements (Steinbrecht et al., 1998).

Results in Fig. 10 also reveal a significant north-south asymmetry in climatological temperature profiles. For instance,

the stratospheric temperatures in the austral spring (e.g., see the October panel) in southern polar region (90°S–80°S) are significantly lower than those in the boreal spring (e.g., see the April panel) in the northern polar region (80°N–90°N), and these temperature differences are associated with lower total $O_3$ columns in the Antarctic than those in the Arctic.

The profile of correlation coefficient in each panel of Fig. 10 illustrates a mutual relationship between $O_3$ concentration and atmospheric temperature. From the upper ($Z^* \gtrsim 35$ km ) stratosphere to the lower mesosphere ($Z^* \lesssim 60$ km), $O_3$ and

temperature are mainly negatively correlated because in this region $O_3$ concentration is mostly governed by photochemical reactions, for which higher temperature speeds up the rate of $O_3$ destruction. From the lower stratosphere down to the upper troposphere (10 km $\lesssim Z^* \lesssim 35$ km) in which $O_3$ concentration is controlled primarily by atmospheric motions (Brasseur and Solomon, 2005), $O_3$ and temperature are positively correlated, since higher $O_3$ concentrations occur likely from adiabatic air parcel compression, which increases the parcel temperature as well, and additionally higher $O_3$ concentrations absorb more UV





**Figure 10.** Climatological temperature profiles corresponding to the M2TCO3 $O_3$ profiles shown in Fig. 6. The color of a solid line indicates the percentage occurrence of the climatological profile, and its line legend displays the average tropopause altitude and the average total column $O_3$. The solid gray line represents the monthly zonal mean temperature profile, and the dashed dark blue line shows the coefficient of correlation between $O_3$ partial pressure and temperature as a function of pressure altitude $Z^*$.





radiations, thus raising the atmospheric temperature. The positive correlation quickly becomes negative as the altitude descends into the lower troposphere ($Z^* \lesssim 10$ km), but swings back and may become positive as the the altitude falls further. In general, the degree of $O_3$-temperature correlation is smaller in the lower troposphere than in the atmosphere above, indicating a weaker connection between them.

The pattern of negative correlation above the upper stratosphere and the positive correlation in the upper troposphere and lower stratosphere between $O_3$ and temperature fields were elucidated with dynamical chemical models (Rood and Douglass, 1985; Froidevaux et al., 1989; Smith, 1995). The $O_3$-temperature correlation profiles in Fig. 10 are consistent with those from numerical modeling (Rood and Douglass, 1985) and observational data analysis (Fortuin and Kelder, 1996), indicating that the MERRA-2 $O_3$ and temperature climatologies represent $O_3$ and temperature distributions and their interrelationship

realistically.

## 4.6   Spatial Distribution of $O_3$

The spatial distribution of $O_3$ is controlled by the various chemical and dynamical processes that drive the production, destruction, and transport of atmospheric $O_3$. Since the distributions of $O_3$ sources and surface topography are inhomogeneous over the globe, systematic patterns with significant longitudinal variations are present in horizontal $O_3$ distribution. The MERRA-2

baseline climatologies presented in previous sections focus on the latitudinal and altitudinal distribution of $O_3$, with a coarse longitudinal resolution retained through the binning of the globe with equal size ($15°$-longitude $\times$ $10°$-latitude) tiles. To better capture the spatial variation of $O_3$ and to compare directly with previous global $O_3$ climatologies (Ziemke et al., 2011; Liu et al., 2013, referred to respectively as Ziemke2011 and GLiu2013 hereafter), we create a higher spatial resolution $O_3$ profile climatology from MERRA-2 by reducing the tile bin size to $5°$-longitude $\times$ $5°$-latitude, which is the bin size for both

Ziemke2011 and GLiu2013 climatologies. This higher spatial resolution MERRA-2 climatology consists of monthly statistics of $O_3$ volume mixing ratio profile, tropopause pressure, stratospheric column $O_3$ (SCO), and tropospheric column $O_3$ (TCO) for 2592 ($= 36 \times 72$) $5° \times 5°$ tiles. Here the SCO is an integration of an $O_3$ profile from 0.01 hPa down to the tropopause, and the TCO from the tropopause down to the surface.

     For better comparisons with the Ziemke2011 climatology, we partition the MERRA-2 total $O_3$ column into SCO and TCO

using the climatological tropopause pressure provided in Ziemke2011, which is compiled from the National Centers for Environmental Prediction (NCEP) tropopause pressures. Figs. 11 and 12 show the spatial distributions of the NCEP-based MERRA-2 SCO and TCO respectively for twelve months of the year, and their comparisons with Ziemke2011. Results in Figs. 11 and 12 reveal excellent agreement between MERRA-2 and Ziemke2011 in the low latitude zone (within $\pm 30°$), with $\Delta$SCO $< \pm 5$ DU and $\Delta$TCO $< \pm 4$ DU for most months and tiles. This agreement becomes worse in the higher latitude regions, mostly

showing larger differences in SCO and TCO between MERRA-2 and Ziemke2011, likely due to higher longitudinal variability in stratospheric $O_3$, degrading the accuracy of SCO from gap-filling interpolation of Aura MLS data (Ziemke et al., 2011). But more significantly, Figs. 11 and 12 show that both spatial distributions of MERRA-2 SCO and TCO and their seasonal cycles closely resemble those of Ziemke2011.

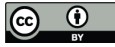


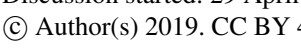

**Figure 11.** Climatological stratospheric column $O_3$ (SCO) maps from MERRA-2 and Ziemke et al. (2011) for the twelve months (January–June, upper panels; July-December, lower panels ) of the year and their differences.

MERRA-2 SCO displays a strong latitude dependence (see Fig. 11) that is shaped primarily by the Brewer-Dobson circulation, showing low values in the tropic ($\sim 220$ DU) and elevated values in the middle and high latitudes, with the highest column amount exceeding 400 DU in the Northern Hemisphere (NH) (between 70°N–80°N) during February–April and exceeding 350 DU in the Southern Hemisphere (SH) (between 45°S–60°S) in September–October. The SCO longitudinal variability is low
5   in the tropic but increases significantly along with higher columns at mid and high latitudes. In the middle-high latitude region centered around 60°N, the longitudinal variation exhibits an oscillatory pattern with high SCO over North America that stretches across the Pacific Ocean to over eastern Asia. This pattern, which is influenced by the semipermanent atmospheric pressure systems resulting from the unique orography distribution in the NH, persists over the year with amplitude varying with





**Figure 12.** Similar to 11, except for tropospheric column $O_3$ (TCO).

the season and reaching its maximum during February. In the SH, a wave-like high SCO pattern with longitude center near the dateline (the 180th meridian) also appears in middle-high latitude band (around $60°$ S), which evolves with the progression of the Antarctic $O_3$ hole from formation to break up, exhibiting SCO build-up and decay outside the stratospheric polar vortex and reaching its maximum in September–October when $O_3$ hole drops to its minimum ($\sim$125 DU).

5    As illustrated in Figs. 11 and 12, TCO behaves differently from SCO, reflecting different sources and dynamical processes that affect its spatiotemporal distribution. The NH mean SCO rises in January, reaches its maximum during February-April, drops in May, and reaches its minimum in September. By comparison, the NH mean TCO starts to increase in March, maximizes in the summer months, and begins to drop in the fall, arriving at its minimum in January-February. These different seasonal cycles indicate a weak relationship between TCO and SCO in NH.





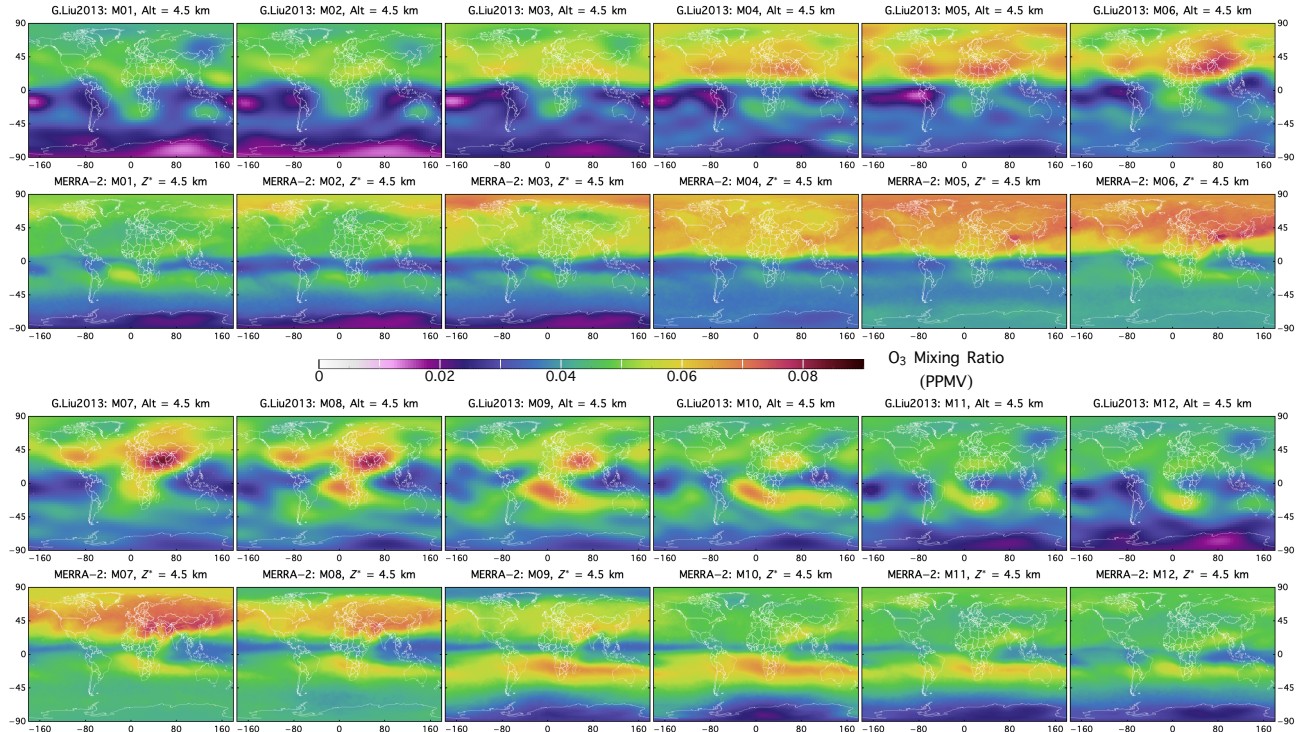

**Figure 13.** MERRA-2 climatological $O_3$ concentration (i.e., mixing ratio in PPMV) at $Z^* = 4.5$ km for the twelve months (January–June, upper panels; July-December, lower panels) of the year and comparison with the tropospheric $O_3$ profile climatology of G. Liu et al. (2013). Note that mixing ratio of G. Liu et al. (2013) is at 4.5 km altitude.

In the tropics, TCO is small (15 to 25 DU) in the Pacific but high (35 to 45 DU) in the Atlantic, exhibiting a stable wave-like pattern that detaches from the SCO distribution, which is almost longitudinally invariant in the same latitude zone. The small TCO in the tropical Pacific is likely due to atmospheric deep convection that uplifts marine boundary air, which is $O_3$ poor, into the middle and upper troposphere. The elevated TCO in the tropical South Atlantic and the connected high TCO band along the edge of southern tropics (30°S) between 40°W and 80°E are contributed from regional $O_3$ productions, including lightning, biomass burning, fossil fuel combustion, and soil emissions, as well as influx from the stratosphere (Thompson et al., 2000; Lelieveld and Dentener, 2000; Moxim and Levy, 2000; Martin et al., 2002; Edwards et al., 2003; Sauvage et al., 2007). This enhancement persists throughout the year, but it is strongest in September-November and weakest in March-May. Its longitudinal variation and seasonal dependence are influenced by large-scale atmospheric circulations (Wang et al., 2006). In the NH extratropics (between 20°N and 40°N), TCO displays a zonal band elevation with strong longitudinal variability and with the highest TCO occurs over the Mediterranean and eastern Asia in June-July. The TCO enhancement in this industrialized zonal band is significantly contributed from anthropogenic emissions (Li et al., 2001; Liu et al., 2011; Jiang et al., 2016).

Aura MLS and OMI $O_3$ measurements are combined to create the Ziemke2011 climatology, and they are assimilated to generate the MERRA-2 $O_3$ field. Consequently, the close resemblance described in this section between MERRA-2 and





Zimeke2011 climatologies are expected, even though MERRA-2 uses a period that is twice as long as that of Zimeke2011. To further evaluate the MERRA-2 climatology, we compare it with the GLiu2013 climatology, which is created from trajectory mapping of long-term ozonesonde record (Liu et al., 2013). Fig. 13 shows monthly maps of $O_3$ concentrations (i.e., mixing ratio in PPMV) at 4.5 km altitude from GLiu2013 and at 4.5 km pressure altitude from MERRA-2 for twelve months of the

year. These maps in Fig. 13 show similar $O_3$ spatial distributions and temporal evolution between the two climatologies. For instance both exhibit persistent $O_3$ concentration enhancements in the NH extratropics and in the tropical and subtropical South Atlantic. However, while GLiu2013 enhancements locate in roughly the same area as MERRA-2, they differ in shapes, likely due to the spatial gaps of ozonesonde data. The seasonal cycles agree well with each other: both show strongest NH enhancements in June-August and weakest in January-March, and strongest SH enhancements in September-November and

weakest in March-May.

The close resemblances of $O_3$ columns (both TCO and SCO) between MERRA-2 and Ziemke2011 and general similarities of $O_3$ concentrations between MERRA-2 and GLiu2013 in terms of their spatial distributions and seasonal cycles demonstrate that the MERRA-2 climatology capture the spatiotemporal $O_3$ distribution realistically.

## 5 Validation

The baseline climatologies constructed from MERRA-2 data describe the systematic behavior of $O_3$ profile and variance and their spatial and temporal dependence on tropopause pressure and total $O_3$ column respectively. Since there is no other $O_3$ profile climatology with a similar resolution and dependency, we validated the MERRA-2 climatologies by downgrading and then comparing them with independent climatological datasets. In section 4.2, we validated the daytime tropopause-dependent (downgraded in LST and longitude) M2TPO3 climatology, which show good agreement with the TpO3 climatology (Sofieva

et al., 2014) compiled from independent $O_3$ profile data. In section 4.6, we validated the spatiotemporal $O_3$ variations represented in downgraded (in LST and $O_3$ columns) daytime M2TCO3 climatology, which shows good agreement with the Ziemke2011 and GLiu2013 climatologies. In this section, we present further comparisons of vertical profiles and integrated quantities (i.e., vertical columns) to demonstrate the validity of M2TCO3 climatology.

We compare the daytime annual zonal mean (i.e., downgraded in LST, season, longitude, and $O_3$ column) M2TCO3 clima-

tology with with the annual zonal mean (downgraded in season) ML climatology (McPeters and Labow, 2012), which is an improved version of the LLM climatology (McPeters et al., 2007). Figure 14 shows the annual mean $O_3$ profiles (upper left and middle panels) and their standard deviations (lower left and middle panels) as functions of altitude and latitude from the daytime M2TCO3 and ML climatologies. This figure also includes the plots of the relative (percent) difference between the mean profiles (upper right panel) and between the standard deviations (lower right panel) from these two climatologies. In the

upper stratosphere ($Z^* > 20$ km), annual mean profiles have an excellent agreement between these two climatologies, with relative differences mostly within $\pm 3\%$ (see Fig. 14 upper panels). However, relative differences are larger but within $\pm 30\%$ in the lower stratosphere and troposphere ($Z^* < 20$ km), mainly due to sampling differences and due to the lack of tropospheric $O_3$ production in the MERRA-2 reanalysis (Wargan et al., 2015; Bosilovich et al., 2015). These differences between M2TCO3





and ML climatologies are within the uncertainties estimated from the differences between two climatologies compiled from different sources, including the ML and the LLM in McPeters and Labow (2012), the tropopause-based $O_3$ (TpO3) climatology and the ML in Sofieva et al. (2014), and the TpO3 and the LLM in Sofieva et al. (2014).

As shown in Fig. 14 lower panels, the standard deviations of the downgraded M2TCO3 climatology are similar to those
5  of the ML climatology, both capture the high variability in the upper troposphere and lower stratosphere (UTLS) and the low variability in the upper stratosphere, with a vast majority of differences within ±30%. The sampling differences contribute to the standard deviation differences exhibited in the lower right panel of Fig. 14.

Figure 15 shows the total $O_3$ columns versus month and latitude from the ML and the downgraded M2TCO3 climatologies and their percent differences, illustrating the good agreements between the two for a vast majority of months and latitude
10  zones, with $O_3$ absolute differences are less than 4%. Absolute differences exceed 4% occur in a few areas only, notably in the northern polar zone for several months during which M2TCO3 total columns are lower than those of ML by more than 4%, and in the SH above 70° latitude for the months from August to December during which M2TCO3 total columns are higher than those of ML by more than 4%. These large biases likely result from spatially inhomogeneous $O_3$ distributions,

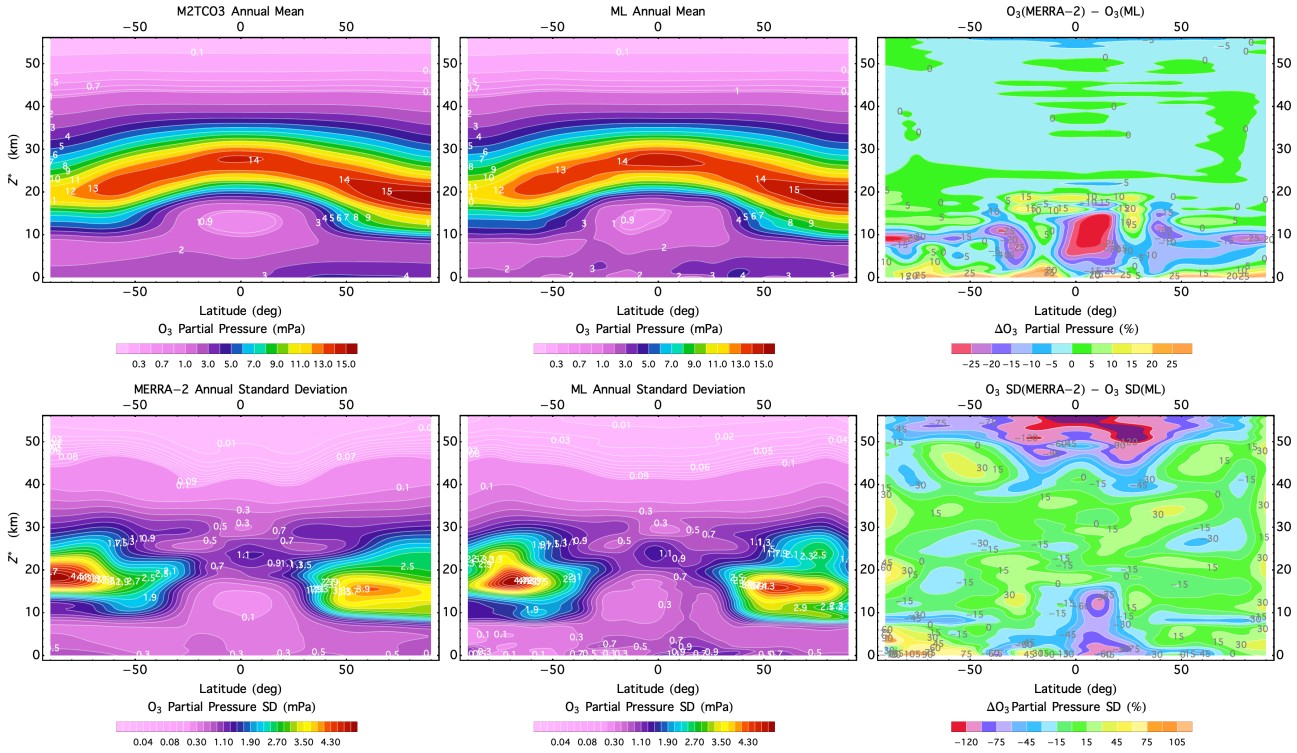

**Figure 14.** Annual zonal mean profiles and standard deviations from M2TCO3 and ML climatologies, and their differences. Upper panels: latitude-dependent $O_3$ partial pressure (in mPa) profiles: M2TCO3 (left), ML (middle), and their relative differences (right). Lower panels: latitude-dependent $O_3$ profile standard deviations (in mPa): M2TCO3 (left), ML (middle), and their percent differences (right).



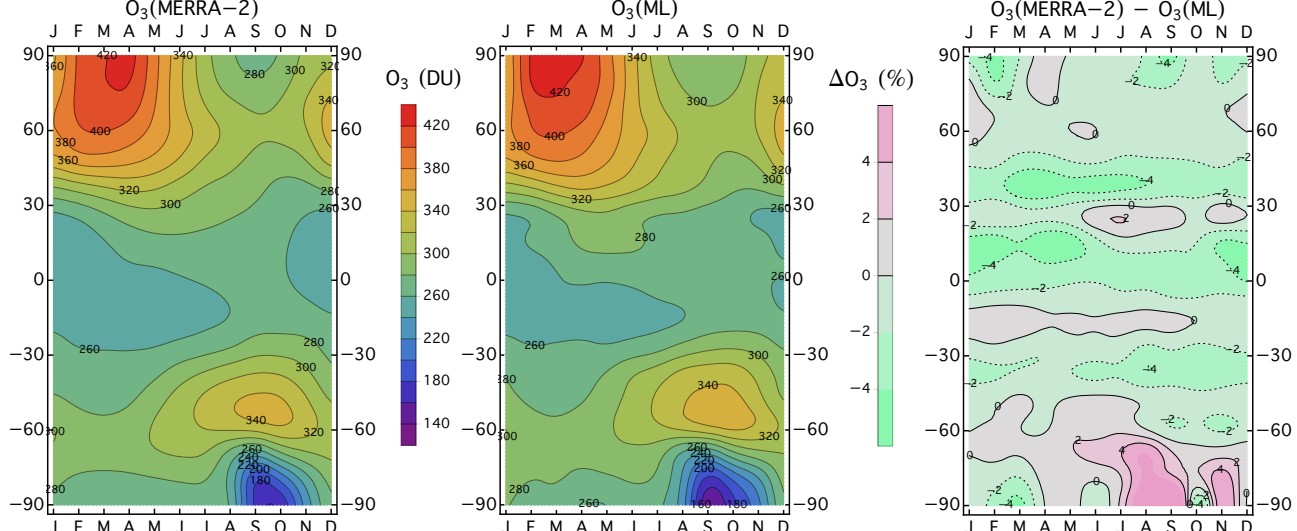

**Figure 15.** Comparison of monthly zonal mean total column $O_3$ from the daytime M2TCO3 (left panel) and the ML (middle panel) climatologies and their relative differences (right panel). Both climatologies capture the strong annual cycle of total $O_3$, which exhibits maximum in late-winter/early-spring in the NH and minimum in late-summer/early-fall in the SH.

such as the longitudinally and seasonally dependent tropospheric $O_3$ distributions (Ziemke et al., 2011; Liu et al., 2013) and the strong $O_3$ gradient across the Antarctic ozone hole, that are sampled differently in creating the two climatologies. While the MERRA-2 assimilation provides a spatially and temporally uniform $O_3$ field, the ozonesonde stations distributed unevenly around the globe provide intermittent $O_3$ profile measurements and the MLS on the polar-orbiting Aura platform samples more

densely at higher latitudes. As a result, the ML climatology, which relies on the ozonesonde data and the Aura MLS $O_3$ profile measurements, samples each latitude zone unevenly, thus contributing to the differences in Fig. 15.

In summary, comparisons of climatological profiles and standard deviations between M2TPO3 and TpO$_3$ (see Figs. 4 and 5), annual zonal mean profiles and standard deviations between M2TCO3 and ML (see Fig. 14), and integrated monthly zonal mean profiles (i.e., the total column $O_3$) between M2TCO3 and ML (see Fig. 15) show good overall agreement, with differences

similar to those between two climatologies constructed from different data sources, thus validates the daytime M2TPO3 and daytime M2TCO3 climatologies.

## 6  Conclusions

Tropopause-pressure-classified (M2TPO3) and total-ozone-column-classified (M2TCO3) climatologies are created from the MERRA-2 $O_3$ profile record between 2005 and 2016, within the period of Aura MLS and OMI assimilation. The enormous

amount of MERRA-2 $O_3$ profile data that cover the globe uniformly and continuously enable precise and accurate representations of systematic behaviors and dynamical variations of $O_3$ vertical distributions. The resulting set of $O_3$ profiles and



covariance matrices capture their dependence on longitude, latitude, local solar time, and season, as well as on tropopause pressure or $O_3$ abundance more accurately over a broader range at a higher resolution than other $O_3$ profile climatologies. Parameterization of $O_3$ profile with tropopause pressure or total column $O_3$ reduces the variability in stratosphere and troposphere compared to the month-latitude dependent climatology, therefore providing improved *a priori* knowledge of $O_3$ vertical

distribution. Both M2TPO3 and M2TCO3 climatologies contain quantitative information about $O_3$ profile covariances, which are not included in previous $O_3$ profile climatologies. The profile covariances provide more realistic constraints on $O_3$ profile retrievals based on the OE inversion technique. Moreover, the EOFs of the climatological covariance matrices facilitate a new scheme to represent the $O_3$ profile and guide a retrieval algorithm to successively improve $O_3$ profile using information contained in spectral measurements.

For profile retrieval algorithms, a closer match between actual and *a priori* $O_3$ profiles, especially in the region where spectral measurements have low vertical resolutions, improves the retrieval accuracy. Thus tropopause-dependent $O_3$ climatology, which reduces the variability further in the UTLS region, is more appropriate for use with $O_3$ profile algorithms. However, the variability reduction is overall higher with column $O_3$ parameterization, indicating more realistic $O_3$ profile assignment based total column abundance. Therefore the column-dependent $O_3$ climatology is uniquely suited for use in total $O_3$ retrieval

algorithms, as the retrieved column determines the likely $O_3$ vertical distribution without needing additional information.

The M2TCO3 climatology provides improved $O_3$ profile representation, capturing systematic profile changes resulting from column variations and their dependence on season and spatial location, which are missing from or insufficiently represented by previous column classified climatologies (e.g. Wellemeyer et al., 1997; Bhartia and Wellemeyer, 2002; Lamsal et al., 2004; Labow et al., 2015) with coarse latitude and time resolutions. The smooth profile change between adjacent dependent variables

(illustrated in Figures in Appendix A) implies that merging profiles from different columns, months, and tiles can provide spatially and temporally continuous representation of $O_3$ profile. The MERRA-2 temperature climatology and the M2TCO3 climatology are used in the $O_3$ and $SO_2$ combo algorithm applied to retrievals from DSCOVR EPIC (EPIC Science Team, 2018), SNPP OMPS-NM (Yang, 2017), and Aura OMI. The description and validation of these $O_3$ and $SO_2$ products will be presented in separate papers.

*Data availability.* The MERRA-2 climatologies are available by contacting the author (kaiyang@umd.edu).

## Appendix A: Figures of the MERRA-2 $O_3$ and Temperature Profile Climatologies

The baseline climatologies consist of statistics of MERRA-2 $O_3$ and temperature profiles collected from 24×18 rectangular tiles at eight different UTC times each separated by three hours. Since each tile has the size of 15° longitude by 10° latitude, the tile statistics at a UTC represent the statistics of an hour of local solar time. Combining the statistics of different tiles within

a range of local solar time downgrades the baseline climatologies to month-latitude (i.e., no distinction in longitude) climatologies, as illustrated in this paper with the example of daytime (9 am – 5 pm) climatologies. Similarly morning, afternoon, or





**Figure A1.** The daytime M2TPO3 climatology contains 2154 $O_3$ profiles that distribute among the 12×18 month-latitude classes. The color of a solid line indicates the percentage occurrence of the profile. The line legends display the average tropopause altitudes and the average total $O_3$ columns. The solid gray line represents the downgraded M2TPO3 profile, i.e., the monthly zonal mean profile.





**Figure A2.** Similar to Fig. A1, except for profile standard deviations in daytime M2TPO3 climatology. This illustrates tropopause pressure classification in general reduces the variability of the climatological profile





**Figure A3.** The daytime M2TCO3 climatology contains 1644 $O_3$ profiles that distribute among the $12 \times 18$ month-latitude classes. The color of a solid line indicates the percentage occurrence of the profile. The line legends display the average tropopause altitudes and the average total $O_3$ columns. The solid gray line represents the downgraded M2TCO3 profile, i.e., the monthly zonal mean profile.



**Figure A4.** Similar to Fig. A3, except for profile standard deviations in daytime M2TCO3 climatology. This illustrates total O$_3$ column classification in general reduces the variability of the climatological profile.





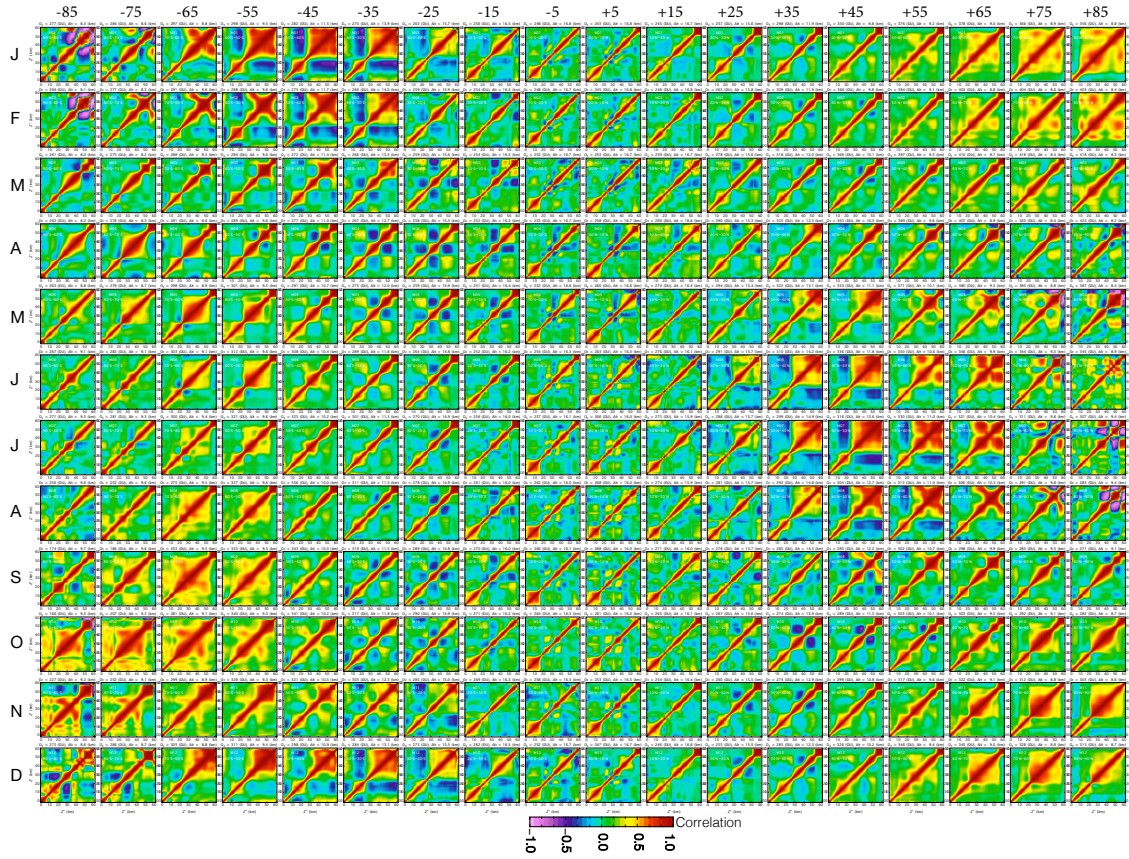

**Figure A5.** O$_3$ profile correlation matrices corresponding to daytime monthly zonal mean profiles. The correlation matrices are standardized or normalized covariance matrices (with 1s in the main diagonal).

nighttime climatologies may be created by selecting the proper local solar time range in combining tiles with the same latitude zone.

In this appendix, each O$_3$ profile in the daytime M2TPO3 and the daytime M2TCO3 climatologies and the corresponding profile variance are displayed in Figs. A1 to A4, which shown O$_3$ partial pressure or standard deviation (in mPa) as a function

5 of pressure altitude $Z^*$ from 0 to 71 km (or 1013.25 hPa to 0.04 hPa). Colored solid lines represent the climatological profiles in Figs. A1 and A3 or the corresponding standard deviations in Figs. A2 and A4. The color of a solid line indicates the percentage occurrence of the climatological profile, and the line legend displays the average tropopause altitude and the average total column O$_3$. The solid gray line represents the downgraded (monthly zonal mean) profile in Figs. A1 and A3 or the corresponding standard deviations in Figs. A2 and A4. Panels in each row show change with latitude, while those in each

10 column reflect seasonal variation.

The MERRA-2 climatologies contain quantitative characterizations of O$_3$ profile covariances. Correlation matrices, which are normalized covariance matrices, associated with daytime monthly zonal means are shown in Fig. A5,





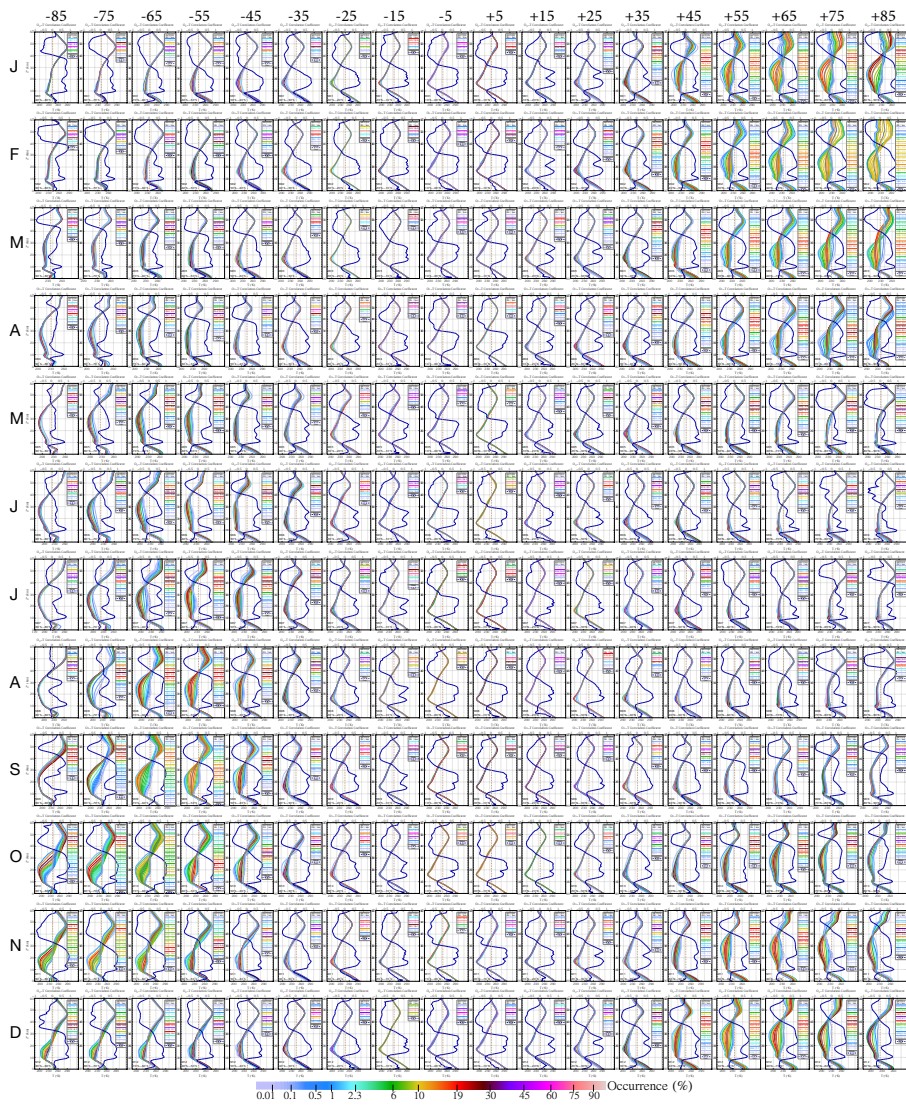

**Figure A6.** Climatological temperature profiles corresponding to the daytime M2TCO3 $O_3$ profiles shown in Fig. A3. The color scheme is the same as that in Fig. A3. The dashed dark blue line represents the coefficient of correlation between $O_3$ partial pressure and temperature as a function of pressure altitude $Z^*$.

Accompanied with the $O_3$ profile climatologies, temperature profile climatologies are created. Figure A6 shows the climatological temperature profiles associated with the daytime M2TCO3 climatology. Figure A6 includes the monthly zonal $O_3$-temperature correlation profiles to illustrate the consistent correlation pattern for all latitude zones and seasons.





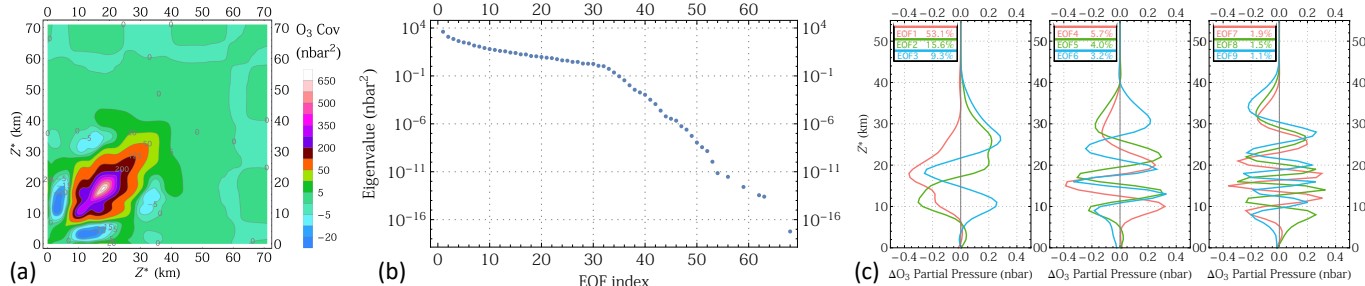

**Figure B1.** (a) The covariance matrix for October and the mid-latitude ($50°$S–$40°$S) zone, (b) eigenvalues of the covariance matrix in (a), and (c) the first nine leading EOFs of the matrix in (a).

## Appendix B: Mapping of a Climatology to a User-Defined Vertical Grid

The baseline climatologies contain $O_3$ concentration (i.e. mixing ratio) profiles specified at $m = 72$ equally spaced pressure altitude ($Z^*$) levels and their covariance matrices. Frequently an application, such as an $O_3$ retrieval algorithm, needs to recast the concentration profiles and their covariance matrices into those of column amount of the vertical layers. In this appendix, we describe the equations for mapping a level-based climatology to user-defined atmospheric layers.

Given an atmospheric layering scheme, it is straightforward to convert a mixing ratio profile $\mathbf{X}$ into a column density profile $\mathbf{\Psi}$ of $n$ layers $\{\Psi_j = X_j A_j,\ j = 1,..,n\}$, where $X_j$ is the mean mixing ratio of the $j^{\text{th}}$ layer, and $A_j$ is the layer air column density, which is determined by the difference between pressures at the layer boundaries.

For a $m \times m$ covariance matrix $\mathbf{S}_m$, which is real and symmetric by definition, hence it may be decomposed exactly as

$$\mathbf{S}_m = \mathbf{Q}\mathbf{\Lambda}\mathbf{Q}^T, \tag{B1}$$

where $\mathbf{Q} = [\mathbf{e}_1, \mathbf{e}_2, \ldots, \mathbf{e}_m]$ is an orthogonal matrix of $m$ columns, with its $i^{\text{th}}$ column $\mathbf{e}_i$ being the $i^{\text{th}}$ eigenvector of the covariance matrix $\mathbf{S}_m$, and $\mathbf{\Lambda} = \mathrm{diag}(\lambda_1, \lambda_2, \ldots, \lambda_m)$ is a diagonal matrix, with its $i^{\text{th}}$ element $\lambda_i$ being the $i^{\text{th}}$ eigenvalue of $\mathbf{S}_m$. Similar to converting mixing ratio profile into layer column density profile, the eigenvectors $\{\mathbf{e}_i,\ i = 1,..,m\}$, also known as the empirical orthogonal functions (EOFs), are converted into layer column density vectors, $\{\mathbf{e}'_i,\ i = 1,..,m\}$, and the elements of the $i^{\text{th}}$ column vector are $\{e'_{ij} = e_{ij} A_j,\ j=1,\ ..,n\}$, where $e_{ij}$ is the mean value of $\mathbf{e}_i$ at layer $j$. The $n \times n$ layer-to-layer covariance matrix $\mathbf{S}_n$ is constructed as

$$\mathbf{S}_n = \mathbf{Q}'\mathbf{\Lambda}\mathbf{Q}'^T, \tag{B2}$$

where $\mathbf{Q}' = [\mathbf{e}'_1, \mathbf{e}'_2, \ldots, \mathbf{e}'_m]$ is a matrix of $m$ column vectors, each has the length of $n$.

We show an example of a covariance matrix $\mathbf{S}_m$ in Fig. B1a, its eigenvalues in Fig. B1b, and the nine leading EOFs in Fig. B1c. The eigenvalues drop rapidly with higher index $i$ (see Fig. B1b), and typically between 15 to 20 leading EOFs account for 99% of the total variances. The $\mathbf{S}_n$ may be reconstructed with a smaller number ($< m$) of leading EOFs in Eq. B2. Doing so may reduce the total variance represented by $\mathbf{S}_n$, thus improve the numerical stability of $O_3$ profile retrievals that use $\mathbf{S}_n$ as an *a priori* constraint.



*Acknowledgements.* The MERRA-2 data used in this study are provided by the Global Modeling and Assimilation Office (GMAO) at NASA Goddard Space Flight Center and are available at the NASA Goddard Earth Sciences (GES) Data and Information Services Center (DISC). This work is supported by NASA.



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
