# Peer review of "Ozone Profile Climatology for Remote Sensing Retrieval Algorithms"

_Atmospheric Measurement Techniques, 2019_

## Referee Comment (RC1) · Anonymous Referee #1 · 3 Jun 2019

This manuscript presents new ozone climatology's constructed based on MERRA-2 analyses with assimilation of AURA MLS and OMI observations during the period 2005-2016. Two types of climatology's have been constructed: one classified as a function of the tropopause height more suited for profile retrievals and the other classified as a function of the total ozone column, which should be used by total ozone retrievals. This work is perfectly suited for AMT and is comprehensively described and illustrated. The general context and past studies are appropriately referred to and discussed. I recommend publication of this work in AMT once the few small comments below have been considered.

**Comments:**

- What is the lower altitude limit of the ozone profiles provided in the climatology's? Do

they extend down to sea level or down to a mean surface altitude? If the latter, how is this lower altitude defined? As the spatial resolution of the climatology is quite coarse compared to current spatial observations, could you provide recommendations on how to adjust the climatological profiles to be used in the retrievals to the actual satellite scene?

- In figure 2, focus is on the diurnal variation of ozone in the upper stratosphere. There must be some diurnal variation in the lowermost troposphere as well, at least in some regions of the world. Can such a diurnal variation be seen in the climatology's? Could you add a comment on this?

- Figures 4-5: It is not fully clear to me what represent the altitudes given in the line legends. As the climatology is binned in 1-km tropopause altitude steps, I would have expected numbers changing regularly (e.g 7.5, 8.5, 9.5 . . . ), which is not the case here.

- Figures 6-7: same question as for Figs 4-5, but this time relating to the O3 column binning. I'd expect a regular grid, like for the TOMS-V8 database. Also could you discuss the O3 column range covered by your climatology? It seems to be significantly smaller for some of the latitude bins comapred to TOMSv8, which might be an issue when using it as an a priori source of information in satellite retrievals. Do you have any recommendation to extend this column range if necessary?

- Figure 15: There is a general bias of 2-4% between the two climatologies, in agreement with Wargan et al. (2017), as mentioned at page 11, line 24. Shouldn't this bias be corrected for before constructing the climatology's or does it have no impact? The mentioned regions in NH where differences would be larger than 4% are not really visible. So this comment is perhaps not necessary.

**Minor/Technical comments:**

- Page 1 – line 18: covraniance –> covariance

- Page 5 – line 24: attitude –> altitude

- Page 5 – Line 32: MEERA-2 –> MERRA-2

- Page 11 – Line 14: tropopuase –> tropopause

- Page 11 – Line 23: a small and high bias? Not clear?

- Page 11 – Line 31: capture –> capturing.

- Page 12 – Line 6: MEERA-2 –> MERRA-2

- Page 15 – Line 10: add "of" –> accuracy of O3 profile retrieval; have –> has.

- Page 19 – Line 19: remove one "the"

- Page 19 – Line 29: The number of 4DU for the TCO differences appears a bit low. There are clearly differences up to 6DU, even at low latitudes. I also see in Fig. 12 that the coverage in latitudes for the maps of the differences is smaller than what the Ziemke2011 database provides. Why is it so? Is it related to the fact that the differences at high latitudes appear to be much higher (a belt of red is visible in Northern hemisphere for the period May-Sept). It seems to come from the Ziemke2011 data and would worth to be mentioned.

- Page 24 – Line 2-3: delete "the tropopause-based O3(TpO3) climatology and the ML in Sofieva et al. (2014),"

- Page 26 – line 2: add a "and" –> "over a broader range and at higher resolution".

- Page 26 – Line 11: 'resolutions" -> "resolution"

- Page 26 – Line 14: add "on" –> "based on total column abundance"

---

## Referee Comment (RC2) · Anonymous Referee #2 · 3 Jul 2019

The climatology described in this paper based on MEERA-2 data is much more complete than a purely data based climatology since observations rarely produce complete sampling. It produces full latitude / longitude coverage. Both total ozone dependent and tropopause dependent versions are created. The negative of course is that it can only be as accurate as the underlying MEERA-2 model. Nevertheless, this climatology is a useful addition for use in remote sensing algorithms. I recommend publication.

---

## Author Comment (AC1) · 11 Jul 2019

We appreciate Referee #1's positive comments and careful review, which improves the manuscript. Below, Referee's comments are in black and our responses in blue.

Referee #1

This manuscript presents new ozone climatology's constructed based on MERRA-2 analyses with assimilation of AURA MLS and OMI observations during the period 2005-2016. Two types of climatology's have been constructed: one classified as a function of the tropopause height more suited for profile retrievals and the other classified as a function of the total ozone column, which should be used by total ozone retrievals. This work is perfectly suited for AMT and is comprehensively described and illustrated. The general context and past studies are appropriately referred to and discussed. I recommend publication of this work in AMT once the few small comments below have been considered.

**Comments:**

- What is the lower altitude limit of the ozone profiles provided in the climatology's? Do they extend down to sea level or down to a mean surface altitude? If the latter, how is this lower altitude defined? As the spatial resolution of the climatology is quite coarse compared to current spatial observations, could you provide recommendations on how to adjust the climatological profiles to be used in the retrievals to the actual satellite scene?

The lower altitude limit of the profile data fields is the sea level (Z*=0 km). The manuscript is revised to describe explicitly the MERRA-2 cliamtologies are sea level based. We also added in Appendix B a recommendation for spatial and temporal interpolation to ensure continuity of *a priori* specification.

- In figure 2, focus is on the diurnal variation of ozone in the upper stratosphere. There must be some diurnal variation in the lowermost troposphere as well, at least in some regions of the world. Can such a diurnal variation be seen in the climatology's? Could you add a comment on this?

The diurnal variation of $O_3$ in the lowermost troposphere can be seen in MERRA-2 data, with its cycle depending on location, season, and altitude above the surface. Since the sources (OMI columns and MLS profiles) of MERRA-2 contain limited information on tropospheric $O_3$, the assimilated data capture only the average behavior of tropospheric $O_3$, but not its variation realistically. The variability of tropospheric $O_3$ profile, including the diurnal cycle, is mostly subdued in the MERRA-2 $O_3$ field. We added to the manuscript some comments and a description of diurnal variation of tropospheric $O_3$.

- Figures 4-5: It is not fully clear to me what represent the altitudes given in the line legends. As the climatology is binned in 1-km tropopause altitude steps, I would have expected numbers changing regularly (e.g 7.5, 8.5, 9.5 . . . ), which is not the case here.

M2TPO3 is binned in 1-km Z* tropopause pressure altitude steps, and the average tropopause altitude (in km) of each bin is displayed in the corresponding legend. This average value does not change regularly from one bin to the next, because the distribution of tropopause altitude is not necessary symmetric respect to the bin center. Furthermore, Z* differs from the actual altitude, contributing to the irregular altitude change from one bin to the next. We added a description in the manuscript to explain the difference of tropopause altitudes displayed in the legends of Fig.4.

- Figures 6-7: same question as for Figs 4-5, but this time relating to the O3 column binning. I'd expect a regular grid, like for the TOMS-V8 database. Also could you discuss the O3 column range covered by your climatology? It seems to be significantly smaller for some of the latitude bins compared to TOMsv8, which might be an issue when using it as an a priori source of information in satellite retrievals. Do you have any recommendation to extend this column range if necessary?

M2TCO3 is binned in 25-DU total $O_3$ steps. Within each bin, the total $O_3$ distribution is usually not symmetric with the bin center. Consequently, the mean total $O_3$ does not change regularly from one bin to the next. On the other hand, TOMS-V8 climatological profiles are created by adjusting the LLM profiles with the TOMS-V8 differential profiles (i.e., the differences between two adjacent standard profiles). This merging is designed to cover preset (latitude-zone dependent) $O_3$ column ranges in 50-DU steps. The preset ranges may be broader than those of M2TCO3 at some months and latitude zones, but they are not found with the MERRA-2 data used to generate the climatology. We added a description in the caption of Fig. 6 to explain the difference of total columns displayed in the legends, and some texts in the manuscript to highlight the range differences.

To expand the total $O_3$ range of M2TCO3 for a tile, linear extrapolation of the column dependent profiles may be sufficient if the extension is not too large (-15 DU < and < 15 DU). Beyond this, one may find the coefficient $\omega_1$ and add $\omega_1 \mathbf{e}_1$ to the profile (see Eq. 1) to generate an M2TCO3 profile with the desired column. We add a description to Appendix B on setting profile to a specific column.

- Figure 15: There is a general bias of 2-4% between the two climatologies, in agreement with Wargan et al. (2017), as mentioned at page 11, line 24. Shouldn't this bias be corrected for before constructing the climatology's or does it have no impact? The mentioned regions in NH where differences would be larger than 4% are not really visible. So this comment is perhaps not necessary.

Evaluation of MERRA-2 by comparing with the SBUV-derived Merged Ozone Dataset (MOD: Frith et al. 2014) concluded that MERRA-2 total $O_3$ column biases low between 0 to –2.3% depending on latitude (Wargan et al. 2017). The source of this error likely comes from the well-known low bias in the OMI total columns, but unlikely from the MLS profiles. Consequently, the vertical distribution of error is inhomogeneous and probably concentrates in the lower troposphere. Bias correction may not improve the accuracy of assimilated profiles without knowledge of error distribution. Therefore, this correction should probably be performed on the OMI data before assimilation.

Reference:

Frith, S. M., N. A. Kramarova, R. S. Stolarski, R. D. McPeters, P. K. Bhartia, and G. J. Labow, 2014: Recent changes in total column ozone based on the SBUV Version 8.6 Merged Ozone Data Set. J. Geophys. Res. Atmos., 119, 9735–9751, doi:10.1002/2014JD021889.

**Minor/Technical comments:**

- Page 1 – line 18: covraniance –> covariance

  Done.

- Page 5 – line 24: attitude –> altitude

  Done.

- Page 5 – Line 32: MEERA-2 –> MERRA-2

  Done.

- Page 11 – Line 14: tropopuase –> tropopause

  Done.

- Page 11 – Line 23: a small and high bias? Not clear?

  Changed to "a slight positive bias"

- Page 11 – Line 31: capture –> capturing.

  Done.

- Page 12 – Line 6: MEERA-2 –> MERRA-2

  Done.

- Page 15 – Line 10: add "of" –> accuracy of O3 profile retrieval; have –> has.

  Done.

- Page 19 – Line 19: remove one "the"

  Done.

Page 19 – Line 29: The number of 4DU for the TCO differences appears a bit low. There are clearly differences up to 6DU, even at low latitudes. I also see in Fig. 12 that the coverage in latitudes for the maps of the differences is smaller than what the Ziemke2011 database provides. Why is it so? Is it related to the fact that the differences at high latitudes appear to be much higher (a belt of red is visible in Northern hemisphere for the period May-Sept). It seems to come from the Ziemke2011 data and would worth to be mentioned.

4 DU changed to 6 DU. We updated Fig.12 to match the latitude coverage of Ziemke2011 TCO, and added the following description to the manuscript: "From September to May, MERRA-2 SCO in the polar latitudes is higher, while TOC is lower than the corresponding columns from Ziemke2011, with the broadest spread occurring in February-April."

- Page 24 – Line 2-3: delete "the tropopause-based O3(TpO3) climatology and the ML in Sofieva et al. (2014),"

Changed to "the TpO3 and the ML".

- Page 26 – line 2: add a "and" –> "over a broader range and at higher resolution".

Done.

- Page 26 – Line 11: "resolutions" -> "resolution"

Done.

- Page 26 – Line 14: add "on" –> "based on total column abundance"

Done.

Referee #2

The climatology described in this paper based on MEERA-2 data is much more complete than a purely data based climatology since observations rarely produce complete sampling. It produces full latitude/longitude coverage. Both total ozone dependent and tropopause dependent versions are created. The negative of course is that it can only be as accurate as the underlying MEERA-2 model. Nevertheless, this climatology is a useful addition for use in remote sensing algorithms. I recommend publication.

We would like to thank Referee #2 for the positive comments. We are mindful of the deficiencies in the MERRA-2 $O_3$ data, and have conducted various profile and column comparisons to validate the new climatologies and to characterize their uncertainties. We will demonstrate improved $O_3$ retrieval accuracy from many satellite obviations using the new climatologies.

---

## Author Response (AR2)

**Associate Editor Decision: Publish subject to technical corrections** (08 Aug 2019) by Mark Weber
Comments to the Author:
Dear Kai and Xiong, you have properly addressed the reviewer's comment so that your paper is accepted after some technical corrections. In your paper you have many multi-panel graphics, which are generally readable (after sufficient zooming in). Figures 8 and 9 could be improved by arranging a) and b) vertically instead of horizontally. This would allow you to make both graphics a bit larger. At the end you should make sure that for the final PDF, all graphics are legible when zooming in (screen display). This is in particular important for the graphics in Appendix A. Congratulation for a very nice paper. Best wishes, Mark

Dear Mark, thank you for accepting our manuscript. As suggested, we have rearranged the panels in figures 8 and 9 to make them larger. Also, we included high-quality figures to ensure decent zoom-in display. We again appreciate your and reviewers' kindness in helping improve the manuscript.
Best.
Kai and Xiong